# Familial Transmission of Developmental Prosopagnosia: New Case Reports from an Extended Family and Identical Twins

**DOI:** 10.3390/brainsci14010049

**Published:** 2024-01-04

**Authors:** Sarah Bate, Ebony Murray, Rachel J. Bennetts

**Affiliations:** 1Department of Psychology, Bournemouth University, Poole BH12 5BB, UK; 2Department of Psychological Sciences, School of Natural and Social Sciences, University of Gloucestershire, Cheltenham GL50 4AZ, UK; emurray4@glos.ac.uk; 3Department of Life Sciences, Brunel University, Uxbridge UB8 3PH, UK; rachel.bennetts@brunel.ac.uk

**Keywords:** prosopagnosia, face recognition, face perception, genetics, heritability

## Abstract

Existing evidence suggests that developmental prosopagnosia (DP) is a surprisingly prevalent condition, with some individuals describing lifelong difficulties with facial identity recognition. Together with case reports of multiple family members with the condition, this evidence suggests that DP is inherited in at least some instances. Here, we offer some novel case series that further support the heritability of the condition. First, we describe five adult siblings who presented to our lab with symptoms of DP. Second, for the first known time in the literature, we describe a pair of adult identical twins who contacted us in the belief that they both experience DP. The condition was confirmed in three of the five siblings (with minor symptoms observed in the remaining two) and in both twins. Supplementary assessments suggested that all individuals also experienced some degree of difficulty with facial identity perception, but that object recognition was preserved. These findings bolster the evidence supporting the heritability of DP and suggest that it can be a specific impairment in some cases.

## 1. Introduction

Developmental prosopagnosia (DP) is a cognitive condition characterised by a relatively selective deficit in face recognition [1,2,3]. DP can be differentiated from the traditionally reported parallel form of the condition, given it occurs in the absence of any neurological injury or illness [1]. Estimates suggest that DP affects many more individuals than acquired prosopagnosia, with some reports suggesting a prevalence rate of 2% of the population [4,5] although this may be an artefact of statistical protocol [6]. Reports of DP in children as young as four years of age imply a lifelong condition in some individuals [4,7,8,9], raising the possibility that DP may sometimes be congenital in origin. This is acknowledged by alternative terminology in some case reports, e.g., congenital [10,11] or hereditary [12,13] prosopagnosia.

In support, DP case reports often describe first-degree relatives who are also suspected [14,15,16] or confirmed [10,13,17,18] to have the condition. Yet, to date, only four detailed family investigations have been reported. Duchaine et al. [18] described 10 members of the same family (both parents, their seven adult children, and a maternal uncle) with deficits in face recognition; Lee et al. [19] described a father and his two adult daughters; Schmalzl et al. [9] found that 7 of 13 relatives across four generations struggled with the recognition of personally familiar faces; and Johnen et al. [20] reported 3 cases of DP within 28 members of the same family, with milder impairments in other relatives. It has been suggested that this pedigree segregation pattern indicates that DP might be of simple autosomal inheritance [12], although specific genes have not yet been identified.

Supporting evidence for familial transmission of face recognition ability per se comes from twin studies in typical perceivers, where correlations between performances on standardised face recognition tasks were nearly doubled in monozygotic compared to dizygotic twins [21]. This was corroborated in a Chinese sample [22] where evidence supporting the heritability of facial identity recognition was also found to extend to tasks tapping configural or holistic processing—the key processing strategy that is thought to underpin face recognition [23,24]. Together with findings of the impressive face discrimination ability in both human newborns [25] and monkeys reared without exposure to faces [26], in addition to early maturity of face-specific processes in children [27,28], this body of work makes a strong case that heritability does occur for face recognition ability [29] while allowing for only a modest (albeit significant) influence of the environment [21,22]. Pertinently, these findings from the typical population suggest that more widespread individual differences in face recognition ability are largely heritable, and add support for the conceptualisation of DP as a continuous rather than a categorical condition (i.e., suggesting that these individuals may simply reside at the bottom of a broad continuum of face recognition ability) [1,6].

Here, we add further support to this emerging body of evidence supporting the heritability of face recognition skills. We offer two new familial case series: a family of five adult siblings who presented to our lab in the belief that they experience DP, and we also describe the first known pair of identical twins with the condition. In all participants, we initially explored whether each individual met the dominant assessment criteria for DP. We then supplemented these measures with tasks of (a) face perception and (b) object processing, in order to provide detailed case studies of each individual and address key theoretical debates in the literature (see below).

## 2. Materials and Methods

### 2.1. Participants

Five white adult siblings (three male; see Figure 1) aged between 55 and 64 years contacted our lab in the belief that they all experience face recognition difficulties. The family suspect that their mother, but not their father, may have also struggled with faces (both now deceased). No further information is known about face recognition ability in their mother’s genealogy. The siblings report a similar and typical upbringing with no known social, emotional, physical, or environmental factors believed to have adversely influenced their face recognition ability. F58 is a financial director and F64 is a teacher. M55 worked as an accountant and then became a teacher, M60 is an accountant, and M62 is a commercial lawyer. No individual reported a history of socio-emotional, psychiatric, or neurological disorder, and all reported normal or corrected-to-normal vision. All five individuals self-reported a high number of prosopagnosia traits on the Prosopagnosia Symptom Checklist (PSC, see Appendix A) [2], and concurrent socio-emotional disorder was also excluded using the Autism Quotient (AQ; scores greater than 32 are indicative of autism spectrum condition, scores in the siblings ranged from 4 to 15, see Appendix A) [30].

One set of white, female identical twins (from an independent family to the siblings described above) aged 22 years also took part in this study. Again, they had reported to our lab with severe everyday face recognition difficulties and also self-reported a high number of prosopagnosic traits on the PSC (see Appendix A). Autism spectrum conditions were excluded in each twin via administration of the AQ (scores were 18 and 24), and neither reported any history of visual, socio-developmental, emotional, neurological, or psychiatric conditions. They were brought up in the same environment, with no other siblings, and no other relatives in their family report face recognition difficulties. Both twins were university students at the time of testing and had engaged in similar social activities and hobbies throughout their lives. Informed consent was collected from all participants, and ethical approval for the study was granted by the institutional ethics committee. Different age-appropriate cohorts of control participants were used as relevant comparison groups in each task. The demographical detail of each cohort is presented in the relevant tables of the Results section and raw data is in the Appendix A.

### 2.2. Materials

#### 2.2.1. Face Memory Tasks

DP is typically confirmed via a self-report of everyday difficulties in face recognition (here represented by self-referral to our lab and completion of the PSC), in conjunction with poor performance on at least two face memory tasks [1,6,31,32]. Thus, we used the Cambridge Face Memory Test (CFMT) [33] and a famous face recognition test [34] to confirm DP in each individual. The CFMT is widely used in prosopagnosia screening [1,6,32,34] and is described in depth elsewhere [33]. In brief, participants are required to learn six target faces and are then asked to recognize those faces across 72 triads with varying levels of difficulty.

Our famous face tests had previously been developed with two age-appropriate versions: one for participants aged 18–34 years and another for those aged 35–65 years [34]. Each version contains the faces of 60 celebrities who were identified to be highly familiar with this age range in the initial pilot testing. Each face is displayed sequentially, in a random order, for an unlimited time period. Participants are asked to type the person’s name or some uniquely identifying biographical fact about that individual. As the measure of interest for prosopagnosia diagnosis is recognition rather than naming [1,6,32,35,36], either a name or a uniquely identifying biographical fact is accepted as a correct response. At the end of the task, participants are provided with a list of the names of the celebrities they had just viewed and are asked to rate their familiarity on a Likert scale that ranges from 1 (not at all familiar) to 5 (very familiar). Any celebrity that is unknown to each participant by name (i.e., those that were scored as a 1 or 2 on the Likert scale) is removed from the overall score and the percentage correct is adjusted accordingly.

#### 2.2.2. Face Perception Tasks

While deficits in face memory are typically used as the hallmark diagnostic indicators of DP [2,37], the overall process of identifying a face is more complex and believed to be composed of multiple sub-processes [38]. As such, neuropsychological investigations into DP individuals have sought to determine whether different aspects of face processing can be selectively impaired [39,40,41,42] or preserved [18,34,41,43,44,45]. Many of these investigations have focused on face perception, where there is mixed evidence that some individuals retain the ability to perceive facial identity where no demands are placed on memory [8,46,47]. However, this evidence is complicated by a lack of confidence in existing face perception tasks, together with concerns about single-test screening for face recognition deficits [48,49,50]. Thus, we supplemented our diagnostic assessments of the individuals using two different face perception paradigms: the Cambridge Face Perception Test (CFPT) [18] and the revised form of the Benton Face Recognition Test (BFRT-r) [47]. 

The CFPT includes 16 trials (eight upright, eight inverted). In each trial, participants are shown a single greyscale target face at the top of the screen and six greyscale test faces beneath, which have been morphed in terms of their similarity to the target face. The task is to sort the test faces in order of their similarity to the target. Participants are allowed 60 s per trial. Performance is measured by the number of errors (i.e., 0 is a perfect score, and 18 is the maximum score for each trial). Error scores are computed by summing the deviations from the correct arrangement for each face (e.g., if a face is two positions from its correct arrangement, two errors are recorded). We calculated error scores for upright and inverted trials separately, converting them to percentage correct using the formula [100 × (1 − (total deviation score/maximum score))] [51]. Further, an inversion effect was also calculated by subtracting the overall accuracy score in the inverted condition from that in the upright condition. Large costs of inversion are often interpreted as evidence for the involvement of configural [52] or holistic [53,54] processing strategies in face recognition. 

The BFRT-r [47] is an updated version of the original BFRT [55,56]. The new task employs naturalistic images of targets that were captured some time apart and, as such, differ in lighting and quality, as well as physical characteristics of the individual (e.g., differences in hairstyle and facial hair). There are a total of 22 trials in which an unfamiliar male target face has to be found among a simultaneously presented array of six male probe faces. In the first six trials, only one of the array faces matched the identity of the target. In the remaining 16 trials, three faces from the array match the target. In the first 12 trials of the task, all faces are displayed from frontal viewpoints. In the final 10 trials, faces are displayed from frontal, but more naturalistic, viewpoints. In each trial, target faces are presented at a slightly different size than those in the array in order to minimise successful matching based on low-level, image-based visual cues. All images are grayscale and display the overall shape of the face, but are cropped below the chin and beyond the hairline. As in the original version of the task, the order of the trials is not randomised and participants have an unlimited length of time to complete each trial. The maximum score on the task is 54: one point for each of the six trials that compose Section 1 (where one response is required per trial), and three points for each of the trials in Section 2 (where three responses are required per trial). The total score was converted to percentage correct, and trial completion times were also measured [47].

#### 2.2.3. Object Recognition Tasks

A core question in the face recognition literature is concerned with the specificity of the skill [57,58,59,60]. Thus, DP case studies are often accompanied by assessments of object recognition, with contrasting findings on preserved [33,57,61,62,63] versus impaired [16,57,61,64,65] skills. Here, we used three tasks to measure object-processing abilities. Two tasks measured object memory: the Cambridge Car Memory Test (CCMT) [66] and the Cambridge Bicycle Memory Test (CBMT) [8]. Both tasks mimic the method and format of the CFMT, using either car or bicycle images, respectively. The CBMT is believed to be of comparable difficulty to its facial equivalent [61].

An additional task measured the object perception ability: the Cambridge Car Perception Test (CCPT) [67]—a test that is identical in format to the CFPT but presents greyscale cars for sorting. The task is scored in the same manner as the CFPT and has previously been used to tap individual differences in object processing within the typical population [67]. However, due to a stimulus error, one upright and one inverted trial was removed from the task.

### 2.3. Procedure

All testing was carried out online using the Testable platform. As a large number of tasks were administered, they were spread over four separate days. On the first day, participants completed the basic DP screening tests in the following order: PSC, AQ, famous faces, and CFMT. On the following three days, participants completed one or two of the perceptual and object tests in different combinations. In all instances, only one memory task was completed per day, which always occurred first.

## 3. Results

### 3.1. Face Memory

For the five siblings, each individual’s score on the CFMT and famous faces test is presented in Table 1, in comparison to the age-matched control data. For the diagnosis of prosopagnosia, a cut-off of two SDs below the control mean has typically been applied to objective face recognition tasks [61,68,69], although some authors argue that this is too conservative when multiple tests are administered and prefer the application of 1.7 SD cut-offs [70,71,72,73]. Very recently, it has been suggested that DP can be partitioned into “major” and “minor” forms of the condition, with DeGutis et al. [32] advising that the diagnosis of DP should follow the rule-of-thumb recommended by DSM-V [74], partitioning the condition into those with major (more than 2 SDs from the mean) versus mild (1-2 SDs from the mean) symptoms. Table 1 illustrates that three of the five siblings met the criteria for “major DP”, achieving scores that fell more than two standard deviations from the control mean on both measures. The remaining two siblings (F58 and F64) also scored below two SDs from the control mean on the famous face task, and between one and two SDs on the CFMT.

The performance of the twins was compared to different age-matched norming data (see Table 2). One twin (T1) reported the use of an effective compensatory strategy (memorising the eyebrows) when completing the CFMT, and subsequently achieved an average score on this task. Given her everyday difficulties with face recognition and her poor performance on the famous face test, we administered an alternative version of the CFMT that uses a different stimulus set (the CFMT-Aus) [48] (for discussion of the need to use multiple screening measures see [49]). Both twins scored more than two SDs from the control mean on this task and on the famous faces test. Thus, given dominant approaches to diagnosing DP require an impaired score on at least two objective scores of face memory, both twins fulfilled this criterion. 

### 3.2. Face Perception

None of the five siblings achieved a low score on the CFPT, all scores in the upright condition were less than one SD below the control mean (see Table 3: all *p*s > 0.20, minimum estimated 22.49% of the control population scoring less than the siblings). The performance was also within the typical range on the inverted and inversion effect measures; inverted: all *p*s > 0.32, minimum estimated 32.51% of the control population scoring less than the siblings; inversion effect: all *p*s > 0.13, minimum estimated 13.03% of the control population scoring less than the siblings. Likewise, no significant difference was observed between the twins and control participants in any measure related to the CFPT; all *p*s > 0.012, minimum estimated percentage of the control population scoring below the twins = 12.14% (see Table 4).

In contrast, accuracy and/or task completion time was significantly atypical in four of the five siblings on the BFRT-r (see Table 3 and Figure 2). Here, we confirmed these effects using more conservative single-case *t*-tests via the SINGLIMS procedure [75], with significance and abnormality of the scores based on one-sided estimates. For accuracy, F64 and M55 both performed significantly worse than the age-matched controls; F64: *p* < 0.001, Z_CC_ = −4.71, estimated percentage of control population scoring below F64 = 0.002%; M55: *p* = 0.03, Z_CC_ = −1.95, estimated percentage of control population scoring below M55 = 3.07%. Completion time was also abnormal in three of the siblings; F58: *p* < 0.001, Z_CC_ = 3.11, estimated percentage of control population with a slower completion than F58 = 0.189%; F64: *p* < 0.001, Z_CC_ = 3.73, estimated percentage of control population with a slower completion than F64 = 0.033%; M62: *p* < 0.001, Z_CC_ = 6.82, estimated percentage of control population with a slower completion than M62 < 0.001%. The remaining sibling, M60, also performed poorly on the BFRT-r, although his accuracy was not significantly poorer than the controls, *p* = 0.063, Z_CC_ = −1.58, estimated percentage of control population scoring below M60 = 6.33%.

Impairments on the BFRT-r were also observed in both twins (see Table 4 and Figure 2). Both T1 and T2 were significantly less accurate and slower than the controls in the BFRT-r; T1 accuracy: *p* = 0.014, Z_CC_ = −2.24, estimated percentage of control population scoring below T1 = 1.40%; T1 completion time: *p* < 0.001, Z_CC_ = 3.42, estimated percentage of control population with a slower completion time = 0.153%; T2 accuracy: *p* = 0.037, Z_CC_ = −1.82, estimated percentage of control population scoring below T1 = 3.68%; T2 completion time: *p* = 0.007, Z_CC_ = 2.48, estimated percentage of control population with a slower completion time = 0.80%. 

### 3.3. Object Recognition

No atypical scores were observed in any of the five siblings on any measure of object processing (see Table 5); CCPT upright: all *p*s > 0.29, minimum estimated 29.25% of the control population scoring less than the siblings; CCPT inverted: all *p*s > 0.400, minimum estimated 40.02% of the control population scoring less than the siblings; CCPT inversion: all *p*’s > 0.140, minimum estimated 13.06% of the control population with a more extreme inversion effect than the siblings; CCMT: all *p*s > 0.182, minimum estimated 18.23% of the control population scoring less than the siblings; CBMT: all *p*s > 0.086, minimum estimated 8.61% of the control population scoring less than the siblings.

Likewise, no impairment in object processing was observed on any measure for the twins (see Table 6). In fact, the twins scored around or above the control mean on all tests of object processing. These findings suggest that object recognition is preserved in all participants, although our use of different control groups for each task prevented us from testing for formal dissociations with face-processing abilities.

### 3.4. Extended Family

Finally, we also had the opportunity to test eight of the five siblings’ 13 adult offspring, but only using the CFMT. Two of these (son of M55, aged 21 years; son of M60, aged 32 years) achieved low face recognition scores (both achieving 42/72) and anecdotally confirmed that they struggle with face recognition in everyday life. These scores fall more than two SDs from the age-matched control mean, suggesting the two individuals might meet formal criteria for DP if they underwent further assessment. The remainder achieved scores in the range of 49–68 correct. None of the offspring that were unavailable for testing are suspected to experience face recognition difficulties.

### 3.5. Summary

In sum, three of the five siblings met the formal criteria for DP, with the remaining two reaching the newly recommended criteria for a “minor” form of the condition (DeGutis et al., 2023). Both twins also reached the threshold for DP diagnosis. Supplementary assessments indicated that four of the five siblings and both twins displayed significant face perception difficulties, with the remaining sibling performing more than 1.5 SDs below average. No sibling or twin was found to have impairments in object recognition. These patterns of face-specific impairments are alternatively segregated according to perception and memory performance on the comparable Cambridge tasks in Figure 3 and Figure 4 for the siblings, and Figure 5 and Figure 6 for the twins.

## 4. Discussion

This investigation reported novel case studies of two families with apparently inherited cases of DP, a group of five adult siblings and a set of identical twins. All participants in both families presented with the hallmark symptoms of DP, with difficulties in familiar and unfamiliar face recognition ability that could not be attributed to any other cognitive, developmental, neurological, or psychiatric condition. 

Evidence for heritability in the family of the siblings is particularly strong; three siblings met the diagnostic criteria for DP and the remaining two presented with a “minor” form of the condition. Further, anecdotal evidence from the siblings suggested that one of their parents was affected, and some objective evidence supported face recognition difficulties in some members of the subsequent generation. There are no clear environmental or occupational factors that could explain why three of the siblings were more affected than others, although it should be noted that the minor cases were observed in the two female siblings and the three males all fell within the “major” DP criteria. However, some caution should be observed in this interpretation, given all the siblings performed very poorly on the famous faces task and the discrepancy between the “minor” and “major” cases was only observed on the CFMT. While this may be attributed to genuine individual differences in face recognition ability, it is equally possible that such small discrepancies in performance can be attributed to measurement error and limitations in the reliability of unfamiliar face memory tasks for single-case diagnostics and comparisons [49]. 

Notably, this investigation is the first to report DP in identical twins. While this further supports the heritability of face recognition ability, it is pertinent that the twins did not report any other family member who is suspected to be affected. Unfortunately, we were not able to confirm this via objective testing, and the case for heritability is also weakened by the consistencies in the developmental environment of the twins where they were both reared within the same home setting and shared the same social interests. Nevertheless, the pattern of findings fits well with previous investigations examining the heritability of typical face recognition abilities within monozygotic versus dizygotic twins [21,22]. Here, heritability was deemed to be the significant factor, allowing for only a modest (albeit significant) influence of the environment [29]. Together, these findings suggest that face recognition across the spectrum may be heritable, supporting the notion of a common continuum of face recognition ability that stretches from very poor through to very good performance.

It should be acknowledged that one twin (T1) performed well within the typical range on one of the key diagnostic tests, the CFMT. This participant achieved a clearly impaired score on the famous faces task, prompting us to administer a further assessment. The CFMT-Aus was designed for this purpose, offering the same paradigm as the original CFMT but with different stimuli. An interview with T1 revealed that she had relied on a particular strategy (memorising the eyebrows) to complete the original CFMT, and this strategy was less successful with the CFMT-Aus. This pattern of findings has not, to our knowledge, been previously reported, and it is noted that the investigation to date has revealed similar control norms for the two versions [48]. Future research might strive to address this specific issue, while perhaps moving beyond the use of such heavily controlled facial images to use those with more a natural variation in appearance that would prevent compensatory techniques from resulting in a successful performance.

In terms of our supplementary assessments, it is notable that at least some impairments in face perception were observed in all DP participants across both families. This is in line with recent suggestions that subtle perceptual deficits may be present in the majority of individuals with DP [39,46,69], in contrast to previous work which proposed a distinction between perceptual and mnemonic cases [5,8,31]. However, in the current sample, perceptual deficits were only detected on the BFRT-r and not the CFPT. Both tasks have been shown to have high sensitivity and predict a unique variance in performance on face recognition tasks [76]; consequently, the difference in performance may reflect the distinctly different paradigms and, therefore, processes tapped by each task. Indeed, two participants displayed impairments in completion time only, reinforcing the importance of measuring both accuracy and speed when assessing impairments in face processing [47]. However, it is acknowledged that different control groups were used to provide norming data for the two tasks in each age group (as was the case for other tasks across the battery), and this may also account for the observed patterns of performance.

No individual showed an abnormal inversion effect, suggesting that configural or holistic processing in this sample was not impaired. This is somewhat unexpected, as previous work has frequently reported reduced or abolished effects of inversion in DP [70]. It is possible that some of the cases in the current study (e.g., F58, T1, T2) have more subtle impairments in holistic or configural processing that do not reach the threshold for significance in single-case comparisons. However, other cases in this study demonstrated relatively typical inversion effects, supporting previous work which has found heterogeneous inversion effects in DP [39,77,78]. As such, it is possible that other perceptual deficits, such as problems in the processing of facial features, may underpin face-processing difficulties in some cases [39,45].

We also observed that object recognition skills were consistently preserved across all the cases we examined. Although one core limitation that prevents us from drawing firmer conclusions is that we were not able to demonstrate these dissociations statistically (this was due to our use of different control groups for each task), scores were within the typical range for all participants on all tasks, and even above the control mean for the twins. One avenue for future research is to develop an object version of the BFRT-r, given this paradigm appeared more sensitive to face perception deficits in the participants reported here and elsewhere [47]. Indeed, preserved performance on such a task would provide more convincing evidence for face-specificity.

Nevertheless, the findings reported here go some way towards suggesting specificity in the heritability of face recognition difficulties, in line with previous studies of typical twins that found very small correlations between the object and face recognition skills of even monozygotic twins [21,22]. This differs from existing familial investigations of DP. Duchaine et al. [18] found that memory for cars and guns was impaired in five of seven siblings with DP, whereas Lee et al. [19] reported impairment in one daughter, with preserved object recognition in the father and another daughter. Thus, consistent with previous findings of heterogeneity in the presentation of DP [6,39,41,79], it seems likely that the specificity of the condition in inherited cases also varies.

## 5. Conclusions

In sum, this investigation presented new familial case studies of DP. Face recognition difficulties were observed in a family of five adult siblings, with three meeting the criteria for DP and the remaining two presenting with more minor difficulties. A pair of adult identical twins also met the criteria for DP, offering the first case report of DP twins to date. Further investigation indicated deficits in face perception in four of the five siblings and in both twins, with no evidence found for object-processing difficulties in any participant. These case presentations add further weight to the heritability of DP and face recognition skills more generally.

## Figures and Tables

**Figure 1 brainsci-14-00049-f001:**
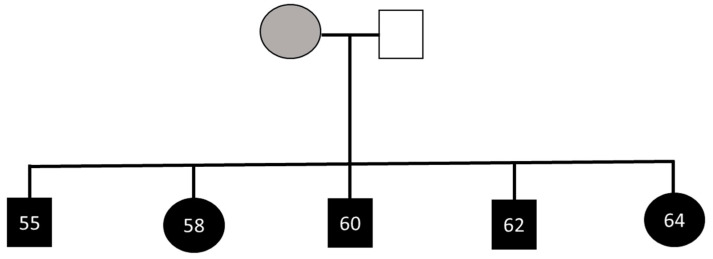
The family pedigree. Black symbols indicate the family members who underwent testing and were found to have face recognition difficulties. The siblings’ mother was suspected to experience face recognition difficulties (shaded grey), whereas their father was not (shaded white). Males are represented by squares and females by circles. Numbers indicate age in years at the time of testing.

**Figure 2 brainsci-14-00049-f002:**
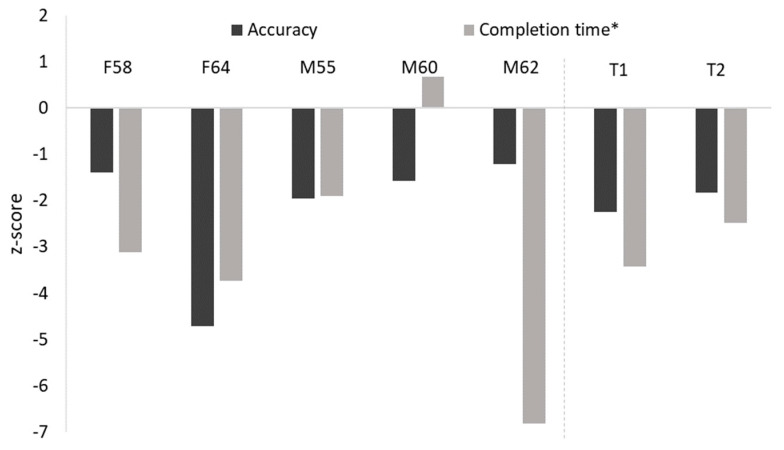
Performance of the five siblings and the twins for accuracy and task completion time on the BFRT-r, displayed as *z*-scores (note: *z*-scores were reversed for the completion time measure, marked with *).

**Figure 3 brainsci-14-00049-f003:**
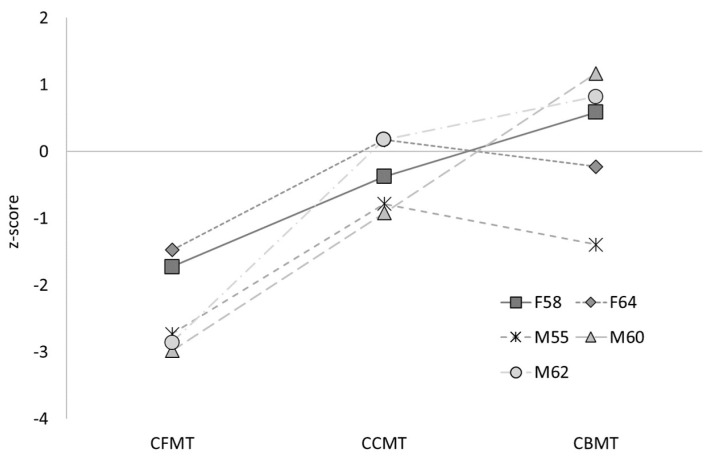
Performance of the five siblings on the face and object versions of the Cambridge memory task, displayed as *z*-scores.

**Figure 4 brainsci-14-00049-f004:**
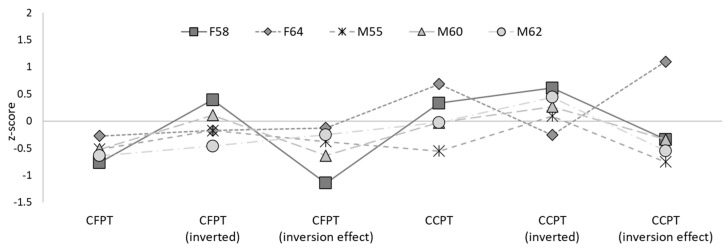
Performance of the five siblings on the CFPT and CCPT, displayed as *z*-scores.

**Figure 5 brainsci-14-00049-f005:**
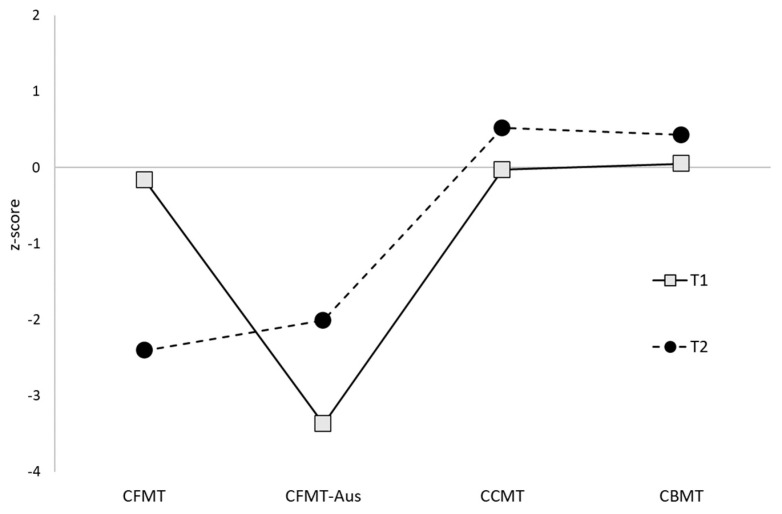
Performance of the twins on the face and object versions of the Cambridge memory task, displayed as *z*-scores.

**Figure 6 brainsci-14-00049-f006:**
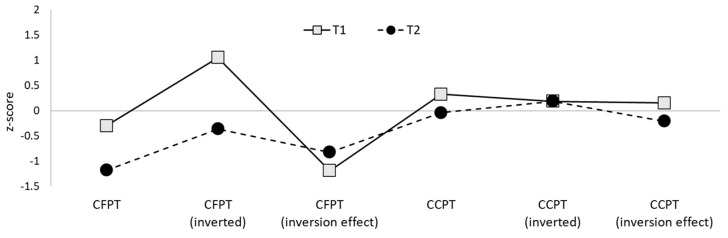
Performance of the twins on the CFPT and CCPT, displayed as *z*-scores.

**Table 1 brainsci-14-00049-t001:** Performance of the family of suspected DP participants on the face memory tasks in comparison to age-matched control norms. CFMT scores are presented both as raw data and percentage correct to aid with diagnostic interpretation given both formats are used in the field. Participant codes in the first column indicate sex (F = female, M = male) and age in years at the time of testing. Age-matched norms for both tasks come from a sample of 47 typical control participants (29 female; M age = 59.9 years, SD = 3.1, range 55–65).

	CFMT Raw Score (% Correct)	Famous Faces (%)
F58	44 (61.11) *	40.43 **
F64	46 (63.89) *	28.57 **
M55	36 (50.00) **	31.67 **
M60	34 (47.22) **	33.33 **
M62	35 (48.61) **	32.14 **
Controls	57.45 (SD = 8.38)	89.97 (SD = 7.76)

* Represents scores that fall 1–2 SDs from the control mean; ** represents scores that fall more than 2 SDs from the control mean.

**Table 2 brainsci-14-00049-t002:** Performance of the twins on the face memory in comparison to age-matched control norms. CFMT scores are presented both as raw data and percentage correct to aid with diagnostic interpretations, given both formats are used in the field.

	CFMT Raw Score (% Correct)	CFMT-Aus Raw Score (% Correct)	Famous Faces (%)
T1	57 (79.17)	33 (45.83) **	39.22 **
T2	38 (52.78) **	43 (59.72) **	57.41 **
Controls	58.43 (SD = 8.03) ^a^	57.73 (SD = 7.34) ^b^	94.37 (SD = 5.58) ^a^

^a^ Age-matched norms come from a sample of 35 typical control participants (22 female; M age = 21.7 years, SD = 1.7, range 18–25); ^b^ norms taken from McKone et al. (2011) [48]: N = 75 (41 female) aged 18–32 years (M = 21.7, SD = 3.0); ** represents scores that fall more than 2SDs from the control mean.

**Table 3 brainsci-14-00049-t003:** Performance of the five siblings and age-matched controls on measures of face perception. Significantly impaired scores are marked with *.

	CFPT % Correct (SD) ^a^			BFRT-r (SD) ^b^	
	Upright	Inverted	Inversion Effect	Accuracy (%)	Task Completion Time (sec)
F58	59.72	55.56	4.16	64.81	859.39 *
F64	65.28	50.00	15.28	31.48 *	951.63 *
M55	62.50	50.00	12.50	59.26 *	679.61
M60	62.50	52.78	9.72	62.96	298.60
M62	61.11	47.22	13.89	66.67	1410.94 *
Controls	68.36 (11.30)	51.71 (9.78)	16.64 (10.91)	78.79 (10.03)	397.98 (148.43)

^a^ Control group: N = 64 (34 female, M age = 58.8 years, SD = 4.4, range = 49–65); ^b^ Controls for the BFRT-r are taken from [47]: N = 42 (23 female; M age = 57.4 years, SD = 1.7, range = 55–60).

**Table 4 brainsci-14-00049-t004:** Performance of the twins on the face memory in comparison to age-matched control norms. Significantly impaired scores are marked with *.

	CFPT % Correct (SD) ^a^			BFRT-r (SD) ^b^	
	Upright	Inverted	Inversion Effect	Accuracy (%)	Task Completion Time (sec)
T1	69.44	63.89	5.55	59.26 *	701.93 *
T2	59.72	50.00	9.72	62.96 *	577.99 *
Controls	72.74 (11.03)	53.55 (9.81)	19.19 (11.46)	78.83 (8.72)	253.75 (130.97)

^a^ Control group: N = 59 (35 female, 1 non-binary, 23 male) aged 20–30 years (M = 24.0 years, SD = 3.2); ^b^ control data taken from [47] (younger control group, Exp 1): N = 93 (49 female), age range 18–30 (M = 24.8 years, SD = 3.4).

**Table 5 brainsci-14-00049-t005:** Performance of the five siblings and matched controls on measures of object processing. There were no significantly impaired scores on any measure.

	CCPT % Correct (SD) ^a^			Object Memory % Correct (SD)	
	Upright	Inverted	Inversion Effect	CCMT ^b^	CBMT ^a^
F58	65.08	68.25	−3.17	63.89	86.11
F64	68.25	60.32	7.94	69.44	76.39
M55	57.14	63.49	−6.35	59.72	62.50
M60	61.90	65.08	−3.17	58.33	93.06
M62	61.90	66.67	−4.76	69.44	88.89
Controls	62.11 (8.98)	62.66 (9.14)	−0.55 (7.72)	67.69 (10.17)	79.12 (11.93)

^a^ Control norms from 61 typical controls (31 female) aged 50–65 years (M = 58.6 years, SD = 4.8); ^b^ Controls for the CCMT: N = 65 (30 male, M age = 58.9 years, SD = 4.4, range = 49–65).

**Table 6 brainsci-14-00049-t006:** Performance of the twins and matched controls on measures of object processing. There were no significantly impaired scores on any measure.

	CCPT % Correct (SD) ^a^			Object Memory % Correct (SD)	
	Upright	Inverted	Inversion Effect	CCMT ^b^	CBMT ^a^
T1	66.67	63.49	3.17	65.28	77.78
T2	63.49	63.49	0.00	72.22	83.33
Controls	63.84 (8.69)	62.02 (7.83)	1.82 (8.74)	65.68% (12.56)	77.05% (14.65)

^a^ Controls: N = 55 (29 female, 25 male, 1 non-binary) aged 20–30 years (M = 25.0, SD = 3.2), ^b^ Controls: N = 59 (35 female, 1 non-binary, 23 male) aged 20–30 years (M = 24.0 years, SD = 3.2).

## Data Availability

The data presented in this study are available in the article and in Appendix A.

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
