# Peer review of "Familial Transmission of Developmental Prosopagnosia: New Case Reports from an Extended Family and Identical Twins"

_brainsci, 2024, doi:10.3390/brainsci14010049_

Round 1

Reviewer 1 Report

Comments and Suggestions for Authors

The manuscript presents new familial case studies of developmental prosopagnosia (DP) that contribute evidence to the case that DP is heritable. The participants were screened for some alternative causes for face recognition difficulties and thoroughly assessed on face and object memory and perception skills, compared against appropriate control groups. The study study was clearly motivated and the results were easy to follow, especially because the behavioral assessments were described such that they could be understood and compared to one another without prior knowledge.

I think the comparison of BFRT-r and CFPT is interesting. Although both perceptual, BFRT-r requires tying the percepts back to an identity, while the CFPT is exclusively focused on perceptual similarity. It's a shame there isn't an object-centric task analogous to the BFRT-r. None of the object tasks required recognizing a specific bike in different settings, times of day, or states of disrepair. Combined with the results you report, if your cases were unimpaired on a task like that, it would be quite compelling that their deficits are face specific.

Author Response

The manuscript presents new familial case studies of developmental prosopagnosia (DP) that contribute evidence to the case that DP is heritable. The participants were screened for some alternative causes for face recognition difficulties and thoroughly assessed on face and object memory and perception skills, compared against appropriate control groups. The study was clearly motivated and the results were easy to follow, especially because the behavioral assessments were described such that they could be understood and compared to one another without prior knowledge.

I think the comparison of BFRT-r and CFPT is interesting. Although both perceptual, BFRT-r requires tying the percepts back to an identity, while the CFPT is exclusively focused on perceptual similarity. It's a shame there isn't an object-centric task analogous to the BFRT-r. None of the object tasks required recognizing a specific bike in different settings, times of day, or states of disrepair. Combined with the results you report, if your cases were unimpaired on a task like that, it would be quite compelling that their deficits are face specific.

Thank you for your positive feedback on our paper. We share your sentiments about the need for an object version of the BFRT-r, and this is under development in our lab but unfortunately is not yet ready for implementation. We recognise that preserved performance on such a task would strengthen the case for face-specific deficits, particularly given the discrepancies in face perception performance between the CFPT and BFRT-r, and have now picked this up in the Discussion (L.426-9).

Reviewer 2 Report

Comments and Suggestions for Authors

Thank you for giving me the opportunity to review this manuscript.

Because this is case series, it is important to describe characteristics and time courses of participants clearly. 

1) Please describe De-identified patient specific information, primary concerns and symptoms of five patient, medical, family, and psycho-social history including relevant genetic information, and relevant past interventions with outcomes clearly.

2) Please describe significant physical examination (PE) and important clinical findings, historical and current information from this episode of care organized as a timeline, diagnostic testing (such as PE, laboratory testing, imaging, surveys), diagnosis (including other diagnoses considered), and prognosis (such as staging in oncology) where applicable, clearly.

3) Please describe Important follow-up diagnostic and other test results, and adverse and unanticipated events.

I think it is necessary to revise the manuscript.

Author Response

Thank you for giving me the opportunity to review this manuscript.

Because this is case series, it is important to describe characteristics and time courses of participants clearly. 

 1) Please describe De-identified patient specific information, primary concerns and symptoms of five patient, medical, family, and psycho-social history including relevant genetic information, and relevant past interventions with outcomes clearly.

 2) Please describe significant physical examination (PE) and important clinical findings, historical and current information from this episode of care organized as a timeline, diagnostic testing (such as PE, laboratory testing, imaging, surveys), diagnosis (including other diagnoses considered), and prognosis (such as staging in oncology) where applicable, clearly.

 3) Please describe Important follow-up diagnostic and other test results, and adverse and unanticipated events.

Thank you for your feedback. As explained in the manuscript, data from the participants was collected online within two sessions that occurred days apart. There are no follow-up sessions or additional information to report. While we appreciate that more detailed physical examinations may be typical for case reports in the medical literature, our paper reports on the condition “developmental prosopagnosia” where online diagnosis and assessment is the primary and typical means of assessment in the field (cognitive psychology rather than medicine). Developmental prosopagnosia is a developmental condition that occurs in the absence of brain injury and other developmental, neuropsychiatric, and cognitive conditions – this is already articulated in the manuscript and we have no further relevant medical or social history to share. The nature of the condition, together with accepted practice in the psychological literature, differentiates our case series from the reporting that is typically observed in medical papers and is in line with existing case reports of developmental prosopagnosia.

Reviewer 3 Report

Comments and Suggestions for Authors

The authors of the manuscript describe unique research material on the study of familial developmental prosopagnosia. On the one side, the number of subjects examined is not large, but on the other side, a sufficient number of valid tests are used.

Typically, in neuropsychiatric practice, states of neurocognitive deficit manifested by prosopagnosia are associated with a specific brain pathology. In this case, due to objective circumstances, it is not possible to make a correlation with diseases.

Overall the manuscript looks good. However, to improve the assessment of this work, I would recommend paying attention to the following:

1. Despite the fact that the authors analyze 79 literary sources, including those from recent years, the Discussion Chapter looks unfinished. A better and more diverse discussion of our own results related to the mechanisms of prosopagnosia can be made. The scientific literature contains information with the simultaneous use of neurophysiological techniques. It may be worthwhile to include classical studies of prosopagnosia in the discussion. For example, L.Szondi?

2. The conclusion should be drawn up separately with a concentration of the most significant results.

3. As the authors present research lines on the assessment of prosopagnosia, the prospects for further research on this condition can be added.

Author Response

The authors of the manuscript describe unique research material on the study of familial developmental prosopagnosia. On the one side, the number of subjects examined is not large, but on the other side, a sufficient number of valid tests are used.

Typically, in neuropsychiatric practice, states of neurocognitive deficit manifested by prosopagnosia are associated with a specific brain pathology. In this case, due to objective circumstances, it is not possible to make a correlation with diseases.

Thank you for your positive feedback. We would like to acknowledge that the reported number of cases of developmental prosopagnosia now outweighs the number of acquired prosopagnosia cases in the literature. The field of developmental prosopagnosia is often considered separately to its acquired equivalent, and it is more frequently considered within broader individual differences in face recognition ability in the typical population.

Overall the manuscript looks good. However, to improve the assessment of this work, I would recommend paying attention to the following:

  1. Despite the fact that the authors analyze 79 literary sources, including those from recent years, the Discussion Chapter looks unfinished. A better and more diverse discussion of our own results related to the mechanisms of prosopagnosia can be made. The scientific literature contains information with the simultaneous use of neurophysiological techniques. It may be worthwhile to include classical studies of prosopagnosia in the discussion. For example, L.Szondi?

Thank you for your suggestion. As explained above we view developmental prosopagnosia more in the context of a broader individual differences debate, as opposed to debating whether it is the equivalent of acquired prosopagnosia. We have added this as context in the Introduction, including a statement on the conceptualisation of developmental prosopagnosia in relation to the acquired form of the condition (L.26-9) as well as broader individual differences (L.56-60). We have now also picked up on this in the Discussion (see L.381-4). Unfortunately a literature search did not return any relevant articles under the author name “L.Szondi”.

  1. The conclusion should be drawn up separately with a concentration of the most significant results.

We have now separated the conclusion from the discussion and added an expanded summary of the findings (L.440-8).

  1. As the authors present research lines on the assessment of prosopagnosia, the prospects for further research on this condition can be added.

Thank you for this suggestion. While this was not a primary aim of our paper we have added some content to the Discussion (L393-6.).

Reviewer 4 Report

Comments and Suggestions for Authors

Bate and colleagues present an interesting case series of two families with developmental prosopagnosia (DP). One family has 5 slightly older (55-64) siblings with DP (2 with mild DP) and the other family has identical twins (22 years old) with DP.  They also demonstrate that DP is rather face specific in both families. This is an important case series to further demonstrate that DP is heritable and specific. There are some remaining issues that would be good to address. 

A major issue is that the authors don't have a control group who have run through their entire battery and instead use various control groups for each task from previous studies. This makes the manuscript lacking in quantitative comparisons and relies more on the reader's ability to qualitatively interpret the pattern of results. A control group run through the identical battery would allow the authors to better quantify the extent to which the members of the families are more similar to each other on the battery than a random number of individuals drawn from the control group (or rather than any 5 individuals with less than average face recognition performance). If the authors have access to a control group (e.g., using prolific.com) to run on this battery, it could substantially improve the manuscript.

There is some concern that twin #1 scored normally on the original CFMT by using eyebrows but scored 24 points lower on the Australian CFMT. This could raise some proverbial eyebrows with readers. Are eyebrows that much less diagnostic on the Australian CFMT? 

Additionally, the way the data is plotted overemphasizes the similarity between siblings. The diagnostic measures (and perhaps the BFRT-r) should be on a different plot than other measures.

Finally, it would be good to pay a bit more attention to potentially differing life experience in individuals in both families. What occupations did the siblings have? Did the twins have any activities (e.g., sports, hobbies, occupations, etc) that differed between them? I understand that the authors may not want to include too much identifying information, but these details are crucial to explaining any potential performance differences observed.

Author Response

Bate and colleagues present an interesting case series of two families with developmental prosopagnosia (DP). One family has 5 slightly older (55-64) siblings with DP (2 with mild DP) and the other family has identical twins (22 years old) with DP.  They also demonstrate that DP is rather face specific in both families. This is an important case series to further demonstrate that DP is heritable and specific. There are some remaining issues that would be good to address. 

 Thank you for your positive comments.

A major issue is that the authors don't have a control group who have run through their entire battery and instead use various control groups for each task from previous studies. This makes the manuscript lacking in quantitative comparisons and relies more on the reader's ability to qualitatively interpret the pattern of results. A control group run through the identical battery would allow the authors to better quantify the extent to which the members of the families are more similar to each other on the battery than a random number of individuals drawn from the control group (or rather than any 5 individuals with less than average face recognition performance). If the authors have access to a control group (e.g., using prolific.com) to run on this battery, it could substantially improve the manuscript.

We are in agreement with the reviewer about this issue, and the only thing preventing us from doing this is budget; unfortunately we have no access to financial resource for this purpose. To collect new norming data on a large number of age-matched participants (different individuals would be needed for the older family members compared to the young adult twins), over such a large number of tasks would unfortunately be costly. We have acknowledged this limitation in the Discussion (L.408-10, L.422-426).

There is some concern that twin #1 scored normally on the original CFMT by using eyebrows but scored 24 points lower on the Australian CFMT. This could raise some proverbial eyebrows with readers. Are eyebrows that much less diagnostic on the Australian CFMT? 

Thank you, we appreciate the point and unfortunately have nothing further to back up the explanation offered to us by the participant, given we are not aware of any specific data that has addressed this issue. We have now added a further paragraph to the Discussion picking up on this issue and recommending further research on the available diagnostic tasks and recommendations for further development (L.385-96).

Additionally, the way the data is plotted overemphasizes the similarity between siblings. The diagnostic measures (and perhaps the BFRT-r) should be on a different plot than other measures.

Thank you - we presented the CFMT scores on the same plots as the CBMT and CCMT given the identical paradigm is used in each test. Scores are presented as z-scores given differences in the difficulty of each task and therefore the norming data. We agree that the famous face task isn’t so relevant to this plot and its inclusion extended the axis unnecessarily. We have therefore removed this task from the plot, but have kept the CFMT in as it gives a direct comparison between the face and object versions of the paradigm (see Figures 3 and 5). We have also separated the perceptual tasks into two separate plots, separating the BFRT-r data  (now Figure 2) from the CFPT/CCPT data (now Figures 4 and 6).

Finally, it would be good to pay a bit more attention to potentially differing life experience in individuals in both families. What occupations did the siblings have? Did the twins have any activities (e.g., sports, hobbies, occupations, etc) that differed between them? I understand that the authors may not want to include too much identifying information, but these details are crucial to explaining any potential performance differences observed.

Thank you for this suggestion – we recontacted the participants and the siblings were happy for us to add their occupations (L.77-9), and the twins were happy for us to add a statement confirming they shared similar social activities and hobbies throughout their lives (L.94-5). We have added some further discussion regarding the siblings’ occupation and whether this could explain differences in the severity of the condition between individuals (L.362-371).

Round 2

Reviewer 2 Report

Comments and Suggestions for Authors

I think that this manuscript was not case series.

Reviewer 3 Report

Comments and Suggestions for Authors

I see work on all my comments. The manuscript can be recommended for publication.

Reviewer 4 Report

Comments and Suggestions for Authors

The authors have sufficiently addressed my concerns.